

# Human-computer interaction based on background knowledge and emotion certainty

Qiang He

State Key Laboratory of Media Convergence and Communication, Communication University of China, Beijing, China

## ABSTRACT

Aiming at the problems of lack of background knowledge and the inconsistent response of robots in the current human-computer interaction system, we proposed a human-computer interaction model based on a knowledge graph ripple network. The model simulated the natural human communication process to realize a more natural and intelligent human-computer interaction system. This study had three contributions: first, the affective friendliness of human-computer interaction was obtained by calculating the affective evaluation value and the emotional measurement of human-computer interaction. Then, the external knowledge graph was introduced as the background knowledge of the robot, and the conversation entity was embedded into the ripple network of the knowledge graph to obtain the potential entity content of interest of the participant. Finally, the robot replies based on emotional friendliness and content friendliness. The experimental results showed that, compared with the comparison models, the emotional friendliness and coherence of robots with background knowledge and emotional measurement effectively improve the response accuracy by 5.5% at least during human-computer interaction.

## INTRODUCTION

Affective computing is an indispensable aspect of harmonious human-computer interaction. Appropriate emotional expression of robots can make humans more receptive to robots with high autonomy and some human-like functions. The personalized emotion generation of robots can also cater to the personalized needs of people with different personalities. Experiments with Vikia robots demonstrate the effectiveness of emotional expression in human-computer interaction (*Bruce, Nourbakhsh & Simmons, 2002*). Artificial emotion models can be divided into emotion models based on rules and evaluation, emotion models based on emotional dimension theory, emotion models based on statistical learning and other types of emotion models. These categories are not completely independent, but have some intersections and connections with each other. The earliest emotion model based on rules and evaluation is the OCC emotion model, which studies the reasoning process of 22 kinds of human emotions and gives the

Corresponding author
Qiang He, heqiang@cuc.edu.cn

corresponding emotion generation rules (*Ortony, Clore & Collins, 2022*). The OCC emotional model provides the basis for many later emotional models, and *Ojha & Williams (2017)* quantified on its basis to form an emotional computing model. The models based on statistical learning include emotion model based on hidden Markov (HMM), artificial emotion model based on neural network, extended finite state machine emotion model and reinforcement learning (*Xin et al., 2013*; *van Kesteren et al., 2000*; *Meng & Wu, 2008*; *Broekens, Jacobs & Jonker, 2015*; *Zhang et al., 2023*). Because the emotional dimension theory is convenient for emotion quantification, most of the artificial emotional models applied to humanoid avatar robots are affective computing models that integrate the emotional dimension theory with the rule evaluation theory, typical ones include Kismet affective model and so on (*Breazeal, 2003*; *Zhang & Ding, 2023*). In addition, there are three dimensional affective models based on PAD (Pleasure, Arousal, Dominance), such as personalized affective models based on PAD (*Yong & Zhiyu, 2012*). There are also some other types of emotional models, such as the emotional interaction model of robots in continuous space (*Kansizoglou et al., 2022*), the emotional model based on Gross's cognitive reappraisal (*Han, Xie & Liu, 2015*), the fuzzy emotional reasoning based on incremental adaptive (*Zhang, Jeong & Lee, 2012*), and the hierarchical autonomy emotional model (*Gómez & Ríos-insua, 2017*). In addition, the release of robots Pepper (*Pandey & Gelin, 2018*) from Japan and Sophia (*Rocha, 2017*) from Hansen Company in the United States has caused a sensation in the field of human-computer interaction, but the specific mechanism of emotion generation is not well understood. Most of these models are designed for specific platforms or rules and lack portability and expansibility, as shown in literature (*Bruce, Nourbakhsh & Simmons, 2002*; *Xin et al., 2013*; *Breazeal, 2003*; *Yong & Zhiyu, 2012*), *etc*. Moreover, many emotional models only stay in the stage of using artificial agents or virtual human simulation, such as literature (*van Kesteren et al., 2000*; *Broekens, Jacobs & Jonker, 2015*; *Yong & Zhiyu, 2012*; *Li et al., 2022*; *Lin et al., 2023*), *etc*.

In this study, by introducing the external knowledge graph (*Yong & Zhiyu, 2012*; *Kansizoglou et al., 2022*) as the background knowledge of robots, this article simulates the awakening process of background knowledge in the process of human communication, and analyzes the emotional friendliness of participants. A human-computer interaction model based on the ripple network of knowledge graph is proposed, aiming at improving the emotional friendliness and coherence of robots in the process of human-computer interaction.

The contributions of this study can be highlighted as follows.

(1) The affective friendliness of human-computer interaction was obtained by calculating the affective evaluation value and the emotional measurement of human-computer interaction.

(2) The external knowledge graph is introduced as the background knowledge of the robot, and the conversation entity is embedded into the ripple network of the knowledge graph to obtain the potential entity content of interest of the participant.

(3) The robot replies based on emotional friendliness and content friendliness.

The following study are organized as follows. In the next section, we briefly review the related work regarding human-robot interaction. Then, we present our method. After that,

we report our experimental results on the 2018 NLPCC task for Open-Domain Question Answering. Finally, we conclude the study.

## RELATED WORK

In recent years, many valuable models of human-robot interaction with emotion and incorporation of background knowledge have been proposed. For example, in *Liu, Xie & Wang (2017)*, a model of continuous cognitive emotion regulation was proposed that can endow the robot with a certain degree of emotional cognitive ability, considering that changes in participant expressions can produce external emotional stimuli for the robot. *Rodríguez, Gutierrez-Garcia & Ramos (2016)* proposed proposed an emotion generation framework that enables robots to produce emotion states consistent with those of the participants in order to achieve intelligent emotion expression, which can lead to emotional trust between the interacting parties during the dialogue. In *Nanty & Gelin (2013)*, the real emotional states in real life are mapped to PAD-3-dimensional space, and the values of each dimension on the three-dimensional number axis are measured by psychological attributes. The PAD emotional space model is used to represent different continuous emotional states of robots. In *Lowe et al. (2015)*, the authors first selected external knowledge of unstructured text related to the dialogue context by term frequency-inverse document frequency (TFIDF) and then used a recurrent neural network (RNN) encoder for knowledge representation, and finally the score of candidate responses was calculated by combining contextual semantics and external knowledge to ensure the accuracy of bot responses. In *Young et al. (2017)*, the triggered long short-term memory (Tri-LSTM) model was used to encode query, response and common knowledge in the external memory module; it was then used to obtain more accurate robot responses so as to solve the problem of lack of background common sense in human-computer interaction. In *Yang et al. (2018)*, a learning framework based on deep neural matching network was proposed, which uses external knowledge to order the response set of the retrieval dialog system, and extracts the knowledge through pseudo-correlation feedback and question and answer, and introduces external knowledge into the deep neural model, which better improves the coherence between contexts in human-computer interaction. *Wang et al. (2021)* gave a comprehensive review of knowledge-enhanced dialogue systems, summarized research progress to solve these challenges and proposes some open issues and research directions. *Pantic & Rothkrantz (2003)* investigated that next-generation HCI designs need to include the essence of emotional intelligence, that is the ability to recognize a user's affective states-in order to become more human-like, more effective, and more efficient.

Although the above studies have to some extent considered the emotional state and external knowledge during human-robot interaction, some of them only consider the impact of a single round of interaction without considering contextual coherence, or only consider the impact of the emotional state in context on the robot's response, or only consider the impact of external knowledge in context on the robot's response. The influence of external knowledge on the robot's responses is also considered. Aiming at the problems of lack of background knowledge and poor reply coherence of robots in the
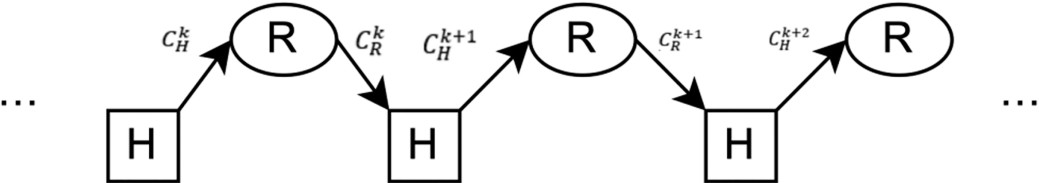

**Figure 1 Input and output in the process of human-computer interaction.**

current human-robot interaction model, this article proposes a human-robot interaction model based on knowledge graph ripple networks by introducing external knowledge graphs as background knowledge of robots, simulating the process of awakening background knowledge during human interaction, as well as analyzing the emotional friendliness of the participants, aiming to improve the emotional friendliness and coherence of the robot in the process of human-robot interaction.

# PROPOSED METHOD

## Problem definition

When two interacting parties engage in a conversation about a certain topic, as the conversation continues, the content related to the topic gradually increases and the background knowledge of both parties is gradually awakened. In the current HCI system, the participant is the one who has the background knowledge in the interaction process, and if the dialogue system does not introduce new content or introduces content that is not relevant enough or not emotionally desired by the participant, it will largely reduce the participant's willingness to have a dialogue and thus lead to the end of the dialogue. In this study, we introduce a knowledge graph as the background knowledge of the interaction system to simulate human-to-human communication, so that the interaction system can take into account the emotional state of the participants and continuously provide emotionally and content-related dialogue content.

The human-computer interaction process is shown in Fig. 1, where H and R represent the participant and robot in the human-computer interaction, respectively. The content input by the participant to the interactive system at the kth time is $C_H^k$, the content of the robot's reply to the participant is $C_R^k$. Let the friendliness of emotional interaction between participant and robot as $R(k)$, if we need to get the kth conversation content $C_R^k$, the specific mathematical expression is

$$f : \left( R(k), \boldsymbol{G}, \boldsymbol{C_H^k} \right) \rightarrow C_R^k \tag{1}$$

Inputs and outputs:

The knowledge graph **G** is introduced at system initialization; during the interaction, the participant input is the kth conversation content $C_H^k$, and the system output is the kth robot's response content $C_R^k$.

In the human-computer interaction process, the following two factors should be considered:

(1) Assessment of the human-computer interaction emotional relationship: human-computer interaction is a continuous interaction process, and the participants' perceptual affection for content affects the continuity of dialogue. Based on this, the interactive friendliness of emotional perception in human-computer dialogue was evaluated first. In the process of interaction, the status of emotional perception of participants showed an overall upward trend, which was considered to be in a good interactive relationship.

(2) Assessment of the relationship between human-computer interaction content: the process of communication between people is a process of awakening background knowledge, and there is a correlation between the contents of communication. Based on this, the relevance assessment is carried out on the content of human-computer dialogue, and the potential interested content of participants is found on the knowledge graph.

## Emotional friendliness

Human-computer interaction is a continuous process, and the current affective state is not only related to the content of the current interaction conversation, but also correlates with the content of the historical interaction sessions. The update function of the HCI affective friendliness $R(k)$ is defined by combining the current and historical interaction sessions as

$$R(k) = \min(1, \max(R(k-1) + W(k) \times C(k), 0)) \tag{2}$$

where $R(k)$ is the kth human-computer interaction emotion friendliness and takes values in the range $[0, 1]$, where a smaller value indicates a worse emotional interaction state and, conversely, a larger value indicates a better emotional interaction state. In particular, an initial value of 0.5 indicates an uncertain human-computer interaction. $W(k)$ is the interaction input affective evaluation value, with an initial value of 0 and a range of $[-1, 1]$. A positive value indicates a positive affective state, while a negative value indicates a negative affective state. The initial value of $C(k)$ is 0, which indicates the reinforcing effect of consecutive positive or negative emotions. This means that when the affective tendencies are the same between the two conversations, the value of $C(k)$ increases and the degree of certainty increases; when the affective tendencies are different between the two conversations, the degree of certainty decreases. In the following, $W(k)$ and $C(k)$ are defined.

### Emotion assessment

In order to better quantify and calculate emotions, the interactive input emotions were quantified as vectors with numerical magnitude according to *Park et al. (2011)*. Within the PAD emotion space, the emotion vector consists of six basic emotion states: happy, surprised, disgusted, angry, fearful, and sad. The definitions are as follows

$$h_l = \left(\boldsymbol{E_p} - \boldsymbol{E_l}\right)\boldsymbol{C_l}\left(\boldsymbol{E_p} - \boldsymbol{E_l}\right)^T, l = 1, 2, 3 \ldots 6 \tag{3}$$

where $\boldsymbol{E_P} = \{p_p, a_p, d_p\}$ denotes the interactive input emotion; $l$ takes the values 1, 2, 3, 4, 5, and 6, representing the six emotions happy, surprised, disgusted, angry, fearful, and sad,

respectively. $E_l$ is the set of coordinates of the basic affective state in the PAD space, and $C_l$ is the set of covariance matrices of the basic affective state in the PAD space; then $h_l$ is the distance between the interaction input affect $E_p$ and the basic affect $E_l$ obtained under the $C_l$ constraint.

We define the interaction input emotion assessment function as $P(E_P)$, as shown in Eq. (5). Equation (4) shows the normalization with respect to emotion intensity $l$. To ensure that the equation is meaningful, specifically $h_l$ is not defined as 0.

$$\begin{cases} p_l = \dfrac{1/h_l}{\sum_{k=1}^{6} 1/h_k}, h_l \neq 0 \\ p_l = 1, \sum_{i=0}^{L} P_i = \sum_{i=l+1}^{6} P_i = 0, h_l = 0 \end{cases} \tag{4}$$

$$P(E_p) = [\boldsymbol{p_1}, \boldsymbol{p_2}, \cdots, \boldsymbol{p_6}] \tag{5}$$

For the kth conversation, the HCI emotion assessment is defined as

$$W(k) = p_1 + 0.6p_2 + 0.2p_3 - 0.2p_4 - 0.6p_5 - p_6 \tag{6}$$

The range of emotion assessment is $[-1, 1]$, and a positive value indicates a positive emotion state, while a negative value indicates a negative emotion state.

### Emotion certainty

Emotion certainty characterizes the effect of a previous emotion state on a later emotion state, which is reinforced when the same emotion occurs consecutively. The dynamics of $C(k)$ is now defined as follows:

$$\begin{cases} C(k) = min\left(1, max\left(0, C(k-1) + \left(1 - \left|\dfrac{W(k) - W(k-1)}{2}\right|\right)\right)\right), W(k)W(k-1) \geq 0 \\ C(k) = min\left(1, max\left(0, C(k-1) + \left(1 - \left|\dfrac{W(k) - W(k-1)}{2}\right|\right)\right)\right), W(k)W(k-1) < 0 \end{cases} \tag{7}$$

Emotion certainty is closely related to the affective assessment of the interaction, with successive positive affective assessments contributing positively to emotion certainty and making the interaction more positive, and successive negative emotion assessments contributing negatively to emotion certainty and making the interaction more negative. The process of assessing emotion certainty is more in line with the psychological process of actual human interaction.

## KNOWLEDGE GRAPH RIPPLE NETWORK INTERACTION MODEL

### Knowledge graph ripple networks

A knowledge graph implements a knowledge mapping of the objective world from a string description to a structured semantic description, describing the rich facts and connections between entities in the objective world in the form of a graph structure. A generic representation of the knowledge graph is a triad, *i.e.*, $G = (H, R, T)$, where

$H = \{e_1, e_2, \ldots, e_N\}$ is the set of head entities in the knowledge base and $N$ is the number of entities; $R = \{r_1, r_2, \cdots, r_M\}$ denotes the set of relationships in the knowledge base and $M$ is the number of entity relationships; $T$ denotes the set of tail entities in the knowledge base and $T \subseteq H \times R \times H$. $H$(head), $R$ (relationship), $T$ (tail) form a triad, *i.e.*, head entity-relationship-tail entity.

In order to mine the potential content of interest to the participants using the knowledge graph, entity linking (*De Cao et al., 2022*; *Yang & Liao, 2022*) is used to extract and disambiguate the content of the participants' conversations to obtain the set of conversational entities during the interaction, and then the set of entities is embedded into the knowledge graph ripple network. The relevant sets in the knowledge graph ripple network are defined as follows.

Based on the propagation model of the knowledge graph ripple network, the set of participant dialogue entities obtained in the kth dialogue is defined as

$$H^k = \{h^k | h^k \in G\} \tag{8}$$

where $G$ is a known knowledge graph, $h^k$ is a dialogue entity, and $k$ denotes the number of dialogue rounds. The triadic ripple set for the set $H^k$ of obtained participant dialogue entities is defined as

$$S_n^k = \{(h, r, t) | (h, r, t) \in G | h \in H^k\}, n = 1, 2, \ldots, N \tag{9}$$

where $n$ indicates the entity at which level of association. For example, $S_1^1$ denotes the level 1 associated entity of the dialogue entity in dialogue 1.

The participant's response set is then obtained through the retrieval formula, and the content of the response set is sequentially passed through the *word2Vec* method (*Styawati et al., 2022*) and the *EmbeddingAverage* vector averaging method (*Chu et al., 2022*) to obtain the sentence feature representation vector $v \in K^d$, where $K$ is the feature representation vector and d is the word vector dimension. The probability $p_i$ of association of each triad $(h_i, r_i, t_i)$ with the sentence feature vector $v$ in the level 1 associated entity of the ripple set $S_1^k$ in the kth conversation is calculated as

$$p_i = softmax(v^T R_i h_i) = \frac{exp(v^T R_i h_i)}{\sum\limits_{(h, v, t) \in S_1^k} exp(v^T R h)} \tag{10}$$

where $R_i \in K^{d \times d}$, $h_i \in K^d$ denote the entity vectors of the ripple set $r_i$ and $h_i$ respectively. The entity vectors of the ripple set triple $(h_i, r_i, t_i)$ are obtained from the knowledge graph feature learning method TransD (*Liu, Xie & Wang, 2017*), corresponding to the entity vectors denoted as $h_i$, $R_i$ and $t_i$. The association probability $p_i$ can be viewed as the probability of measuring the similarity of a word vector $v$ to an entity $h_i$ in the relational $R_i$-space.

After obtaining the correlation probabilities, the influence of the dialogue entity on the level 1 associated entity tail entity $r$ is then calculated and the vector $o^1$ is calculated as

$$o^1 = \sum_{(h_i, r_i, t_i) \in S_1^1} p_i t \tag{11}$$

The vector $o^1$ is the 1st order response of the sentence feature vector $v$, where $t_i \in K^d$. Through Eqs. (10) and (11), the content of interest to the participant is propagated farther along the hierarchical relationship of the knowledge graph.

Replacing the sentence feature vector $v$ in Eq. (10) with the vector $o^1$ in Eq. (11), we obtain the 2nd order response $o^2$, which is performed iteratively over the ripple set, *i.e.*, the case where the values of $n$ in $S_n^k$ are 1, 2, respectively, representing the 2nd order propagation of the participant dialogue entity over the knowledge graph ripple network, because too large an order response dilutes the useful information, so the value of $N$ in Eq. (9) is either 2 or 3. In summary the participant's response equation with respect to the vector $v$ has:

$$O = \sum_{i=1}^{H} o^i \tag{12}$$

The normalized content probabilities are

$$y = \sigma(O^T v) \tag{13}$$

Define Eq. (13) as the participant content friendliness, where the sigmoid function is

$$\sigma(x) = \frac{1}{1 + exp(-x)} \tag{14}$$

Combining Eqs. (2) and (13), the content friendliness and sentiment friendliness weights are added to the candidate response content and normalized as

$$y_v = \alpha R(k) + \beta y \tag{15}$$

where $\alpha, \beta$ is the constraint factor and has $\alpha + \beta = 1$ (the default will be $\alpha = \beta = 0.5$, as detailed later in the experimental discussion section). $y_v$ takes on a value range of $[0, 1]$, with values closer to 1 indicating that the participant is more satisfied with the response.

### Knowledge graph ripple network interaction model construction

Firstly, the knowledge graph $G$ is known (this article uses the Chinese general encyclopedic knowledge graph CN-Dbpedia, which is developed and maintained by the Knowledge Workshop Laboratory of Fudan University), the input of this article's model is the $k$th participant interaction input content $C_H^k$, and the output is the $k$th$+l(l > 1)$ robot's reply content $C_R^{k+1}$; then, the continuous emotional state vector of the participant is obtained through the emotion friendliness Then, the continuous emotional state vector of the participant is obtained through the knowledge graph ripple network, while the content friendliness of the participant's conversation is obtained through the knowledge graph

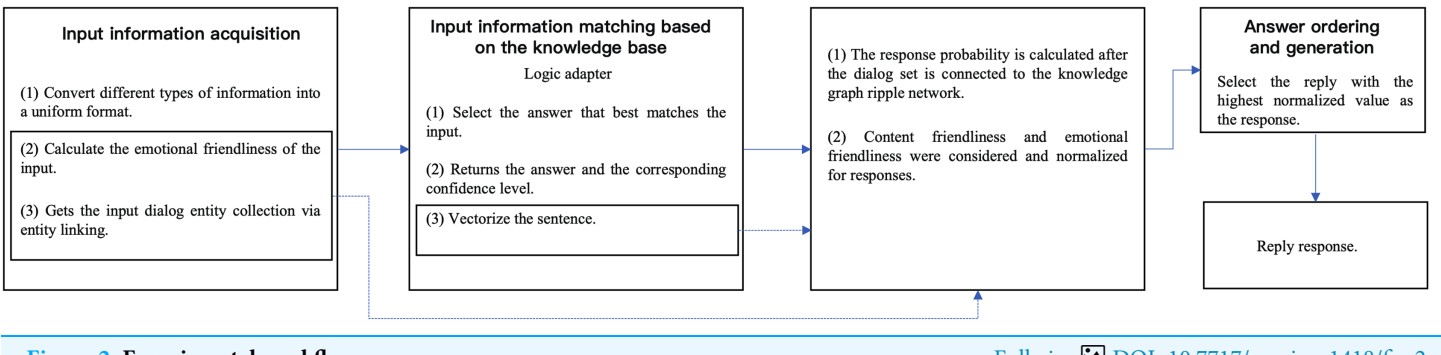

**Figure 2  Experimental workflow.**

ripple network; finally, the optimal reply of the robot for the conversation is given by considering the participant's emotional state vector and the content friendliness together.

In each round of human-computer interaction, the emotional friendliness is obtained by matrix operations between the set of coordinates and positions in the emotional state space and the set of covariance matrices, and its time complexity is of constant order $O(1)$; in the optimal entity content selection based on the knowledge graph ripple network, the number of dialogue content entities is of constant order and the number of entities involved in ripple propagation is also of constant order, then the ripple network propagation time complexity is of constant order $O(1)$; In the case of optimal entity content selection based on knowledge graph ripple networks, the number of dialogue content entities is constant and the number of entities involved in ripple propagation is also constant, then the time complexity of ripple network propagation is of constant order $O(1)$. Considering that the number of rounds of human-computer interaction is $n$, then the time complexity of this model is $O(n)$.

# EXPERIMENTAL STUDIES

## Settings

In order to conduct an effective experimental comparison of the HCI model proposed in this study, ChatterBot in Python is extended to the model proposed in this study and simulated in the form of a text chat. Figure 2 shows the workflow of the model in this study, where the solid part is the ChatterBot framework and the dashed part is the extension. Emotional friendliness is calculated by gestures, speech and facial expressions during human-computer interaction.

The used data were obtained from the dialogue *corpus* of the 2018 NLPCC task for Open Domain Question Answering, which is a Chinese dialogue question and answer *corpus*. The *corpus* has a total of 24,479 question-answer pairs, and 2,500 question-answer pairs are randomly selected as the validation set, another 2,500 question-answer pairs are randomly selected as the test set, and the remaining question-answer pairs are used as the model training set.

In this study, the following four models are selected for comparison.

(1) In *Sutskever, Vinyals & Le (2014)*, the proposed model is a dialogue model for automatic response generation based on Seq2Seq (Sequence to Sequence) of LSTM (Long Short-Term Memory).

(2) In *Gunther (2019)*, the proposed is a ChatterBot interaction model with answer ranking output based on confidence level.

(3) In *Zhang, Wang & Mai (2018)*, the proposed is a cognitive model of dialogue with emotional MECs that takes into account the "empathy" of the participants during the interaction and selects emotionally similar responses as answers.

(4) In *Lowe et al. (2015)*, the proposed is a ConceptNet cognitive model that stores general knowledge in an external memory module and integrates relevant general knowledge into a retrieval-based dialogue.

## Evaluation metrics

The mean reciprocal rank (MRR) and mean average precision (MAP) were used to objectively evaluate the model response accuracy, where MRR reflects overall accuracy and MAP reflects single value accuracy, which are defined as

$$MRR = \frac{1}{|k|} \sum_{i=1}^{k} \frac{1}{rank_i^q} \quad (16)$$

$$MAP = \frac{1}{|k|} \sum_{i=1}^{k} Ave(A_i) \quad (17)$$

$$A_{ve}(A_i) = \frac{\sum_{j=1}^{n} (r(j)/p(j))}{n} \quad (18)$$

where, $k$ represents the number of conversations that participants participate in, $rank_i^q$ represents the rank of the i-th participant's reply in the reply set, $A_{ve}(A_i)$ represents the average accuracy of the reply sort of the i-th conversation model. $p(j)$ represents the ranking level of the $j$-th answer in the adjusted reply set after the model adjusts the ordering of the candidate reply set considering the given limiting factors. $r(j)$ represents the ranking of the j-th answer in the standard response set. $n$ indicates the number of replies in the standard reply set. MAP reflects the average accuracy of the one-value accuracy of the recovery performance.

In order to further verify the effectiveness of the model, a manual evaluation method was adopted, and 40 volunteers were recruited according to the requirements to conduct interactive conversations with different cognitive model chatters. Firstly, the validity of each model is measured from the time dimension, and the duration of human-computer interaction is calculated. The volunteers then scored the fluency and sentiment evaluation criteria shown in Table 1.

## Experimental results

In order to calculate the results of MRR and MAP two objective evaluation indexes, the number of standard response set was set as $n = 10$, that is, there were 10 candidate

| Table 1 Fluency and Sentiment evaluation criteria. | |
| --- | --- |
| **Fluency** | **Evaluation criteria** |
| +2 | Content is relevant, grammar is smooth, in line with human communication |
| +1 | Content logic barely related, syntax expression is OK |
| +0 | The content logic is not relevant, the answer is not the question, the expression is confused |
| **Sentiment** | **Evaluation criteria** |
| +2 | The response is emotionally appropriate and expressive |
| +1 | Respond emotionally appropriately |
| +0 | The expression is vague and meaningless |

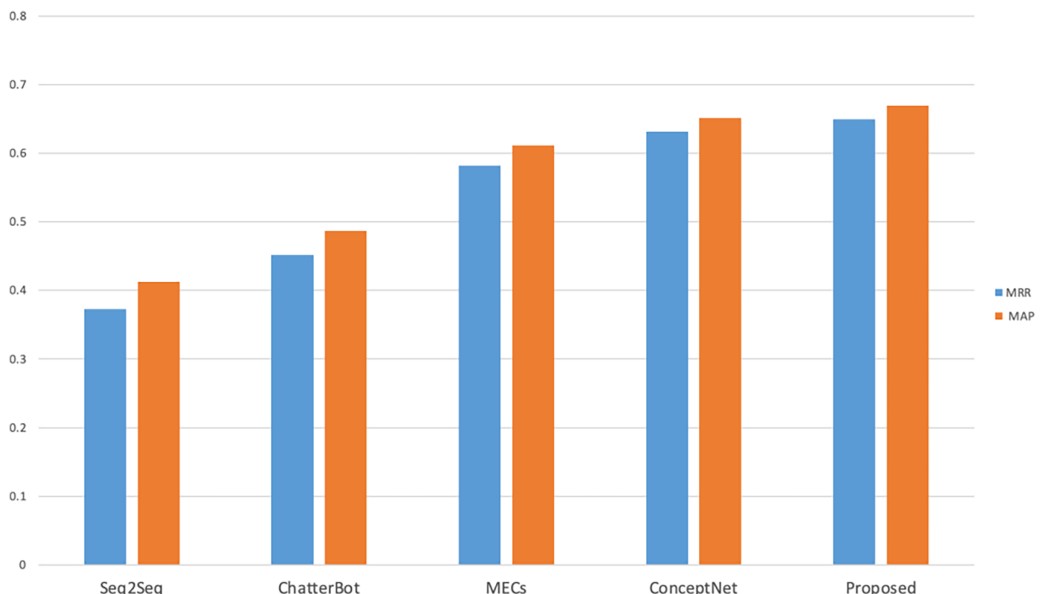

**Figure 3 Objective evaluation results of different cognitive models.**

responses in the standard response set, and the calculation results were shown in Fig. 3. As can be seen from Fig. 3, compared with the other four comparison models, the proposed model achieves better results, mainly because the model proposed in this study considers both sentiment friendliness and content friendliness when ranking the candidate response set, which not only constrains the response content in terms of subjective sentiment friendliness, but also in terms of objective entity coherence. The MECs and ConceptNet models achieved better results when comparing the Seq2Seq and Chat-terBot models, as the MECs considered empathy in the interaction process and ConceptNet introduced external knowledge graphs as common-sense knowledge, both of which considered the sentiment and background knowledge factors respectively. The lowest Seq2Seq scores were due to the fact that many meaningless responses were generated during the interaction, taking into account factors such as security. The objective evaluation verifies that the model is effective in improving the response accuracy.
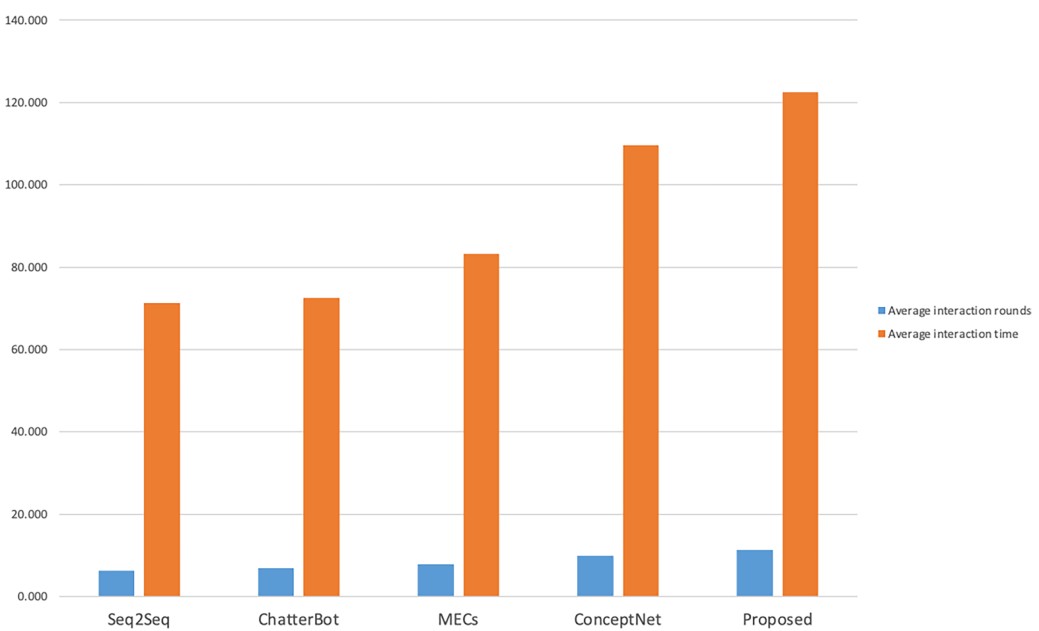

**Figure 4  Statistics of interaction rounds and time.**     

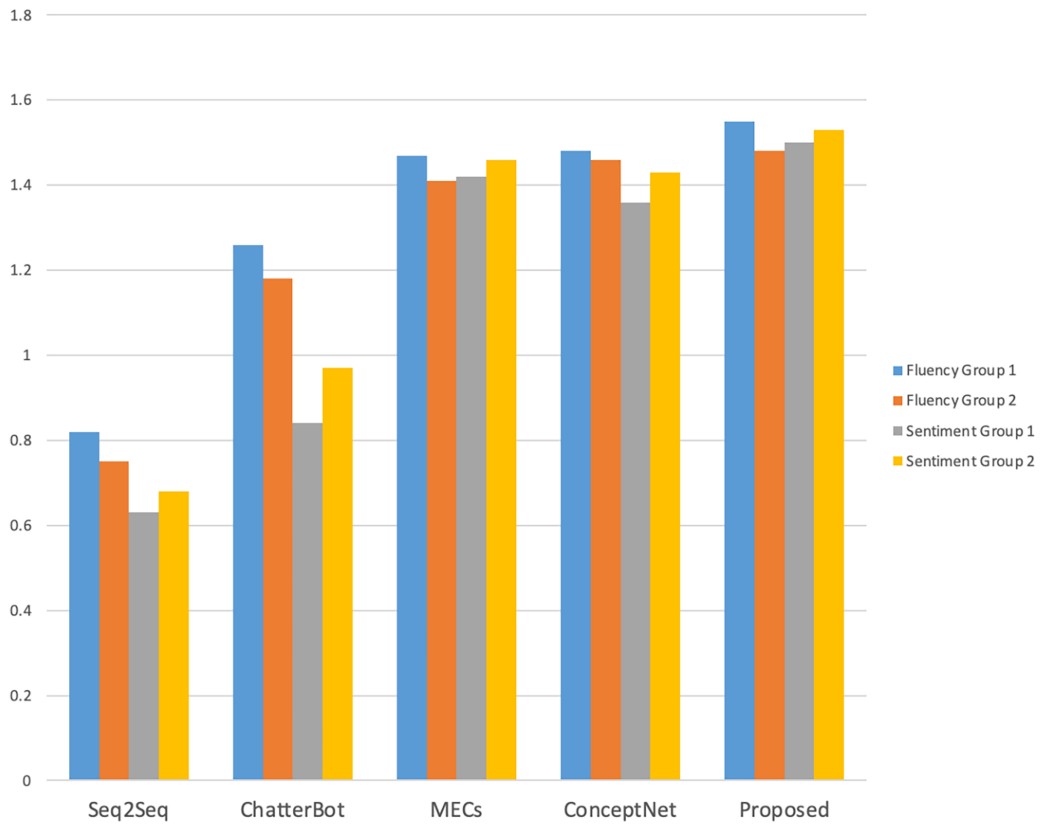

**Figure 5  Fluency and sentiment of the male and female volunteers.**

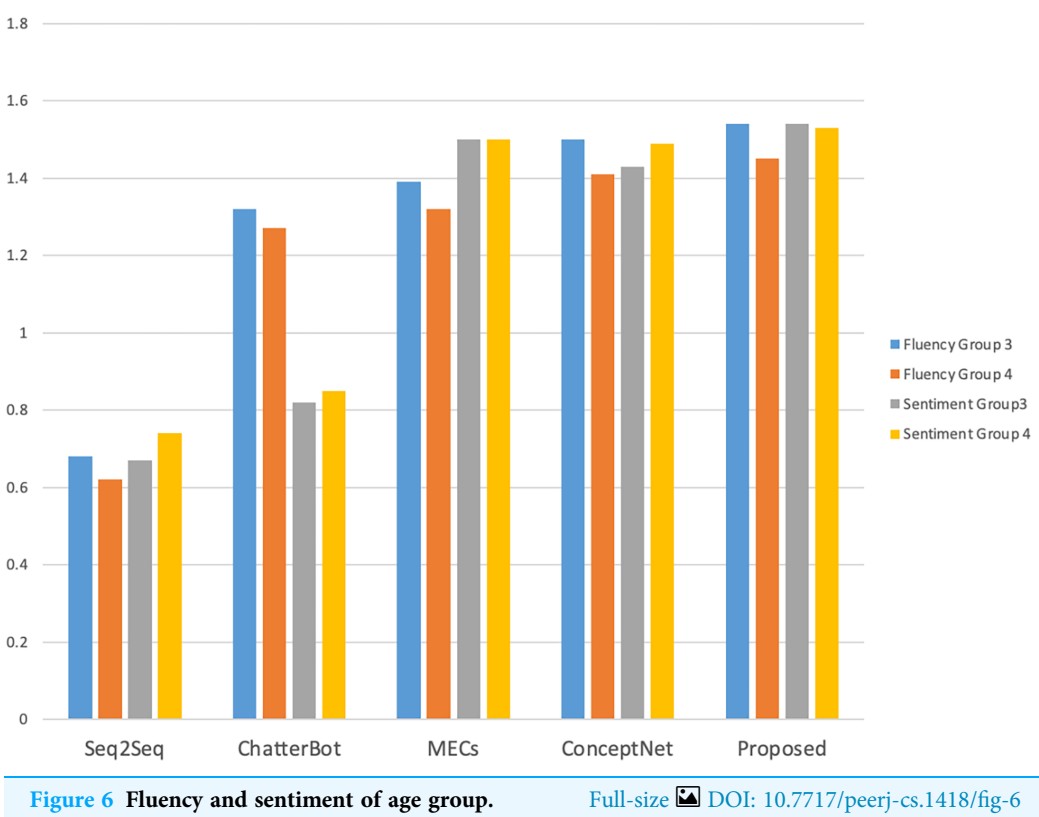

**Figure 6 Fluency and sentiment of age group.**

The manual evaluation took into account the influence of different age and gender groups on the evaluation of the interaction effect of the dialogue system. A total of 40 volunteers of different ages and genders were invited to interact with each model, and the number of rounds of interaction and the interaction time were counted. The results are shown in Fig. 4. A total of 20 male and 20 female volunteers were in each gender group (Group 1 and Group 2 respectively), and two female and two male volunteers were in each age group (19–22 years old for undergraduate students, 23–25 years old for master students, and 26–28 years old for social workers).

As shown in Fig. 4, compared with other models, volunteers performed better with this model in terms of the number of interaction rounds and interaction time; as shown in Figs. 5 and 6, volunteers in different gender and age groups scored this model higher than other models in Fluency and Sentiment. The human evaluation verifies that this model can effectively extend the number of rounds and duration of human-computer interaction by combining content coherence and emotional friendliness.

## Experimental discussion

To further analyze the actual impact of sentiment friendliness and content friendliness on the model, a discussion of Eq. (15), *i.e.*, the constraint a in $y_v = \alpha R(k) + \beta_y$, is carried out, as there is a relationship of $\beta = 1 - \alpha$, so only the impact of taking values of $\alpha$ between [0, 1] on the model needs to be discussed. The objective measures of MRR and MAP are

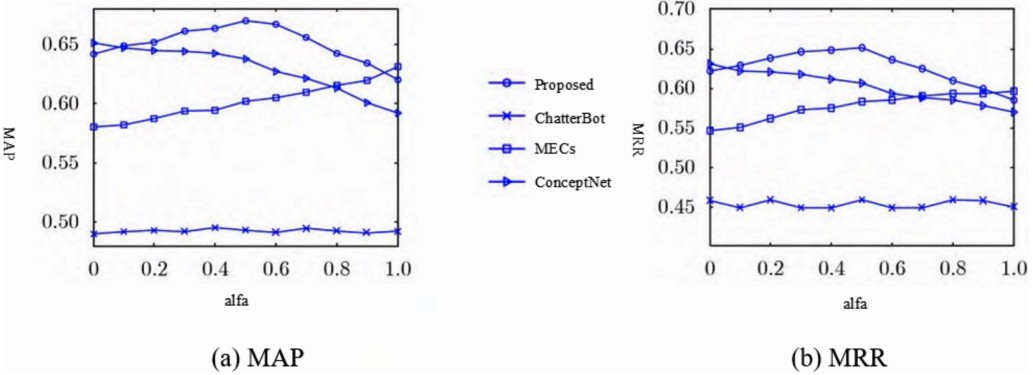

**Figure 7 Automatic evaluation results of different models in terms of (A) MAP and (B) MRR.**

still used to calculate the two indicators as a measure, and again the number of standard response sets $n = 10$ is taken to calculate the two indicator values.

As can be seen from Figs. 7A and 7B, when $\alpha$ tends to zero, this model achieves better results in the objective evaluation of MAP and MRR compared to the ChatterBot model and the MECs model, but the difference is smaller compared to the Concept-Net model, because both this model and the ConceptNet model only consider content coherence under this constraint. The reason is that both the model and the ConceptNet model only consider content coherence under this constraint. As $\alpha$ converges to 1, the model achieves better results for MAP and MRR compared to the ChatterBot and ConceptNet models, but less so than the MECs model, because both the model and the MECs model only take into account emotional factors under this constraint. As the value of $\alpha$ converges to 0.5, this model achieves better results in both MAP and MRR objective ratings compared to the comparison models, as it takes into account both emotion and content friendliness, which can effectively improve the response accuracy.

## CONCLUSIONS

In this study, the human-robot interaction model based on knowledge graph ripple networks is proposed to consider the knowledge graph as the background knowledge of the robot, and use the ripple networks to simulate the awakening of locally relevant background knowledge in the process of human-human communication, extract the potentially interesting entities of the participant, and consider the subjective emotional friendliness of the participant to optimally select the dialogue responses of the robot. Comparative experimental results show that the proposed model can effectively improve the emotional friendliness and coherence of the robot during human-robot interaction. The human-robot interaction model in this study simulates the real human-human communication process, and provides a useful exploration for a more natural and intelligent human-robot interaction system.

The extension of the reasonable field of model parameters is also the direction of further research. What is more, the efficiency of HCI algorithms should also be paid more attentions.

## ACKNOWLEDGEMENTS

We would like to thank all reviewers.

### Funding

The author received no funding for this work.

### Competing Interests

The author declares that they have no competing interests.

### Author Contributions

- Qiang He conceived and designed the experiments, performed the experiments, analyzed the data, performed the computation work, prepared figures and/or tables, authored or reviewed drafts of the article, and approved the final draft.

### Data Availability

The raw data are available in the Supplemental Files.

The data is available at 2018 NLPCC task 7: http://tcci.ccf.org.cn/conference/2018/taskdata.php.

### Supplemental Information

Supplemental information for this article can be found online at http://dx.doi.org/10.7717/peerj-cs.1418#supplemental-information.

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
