# Peer review of "Human-computer interaction based on background knowledge and emotion certainty"

_PeerJ Computer Science, doi:10.7717/peerj-cs.1418_

## Round 0.1 · original submission · Major Revisions

Both reviewers found your research to be of some interest and the paper to be reasonably well structured. However, they each gave some constructive comments. For example:
1. The contribution of this study should be highlighted at the end of the Introduction section.
2. In the Related work section, it is better to cite more references and discuss to motivate this study.
3. Please give the direction of subsequent research in the summary section.
Based on the reviewers' comments, I suggest a major revision.

·

Basic reporting

no comment

Experimental design

no comment

Validity of the findings

no comment

Additional comments

The authors proposed a human-computer interaction model based on a knowledge graph ripple network. The model simulates the natural human communication process to realize a more natural and intelligent human-computer interaction system. Overall, this is a well-organized study. Please consider the following comments which may improve the quality of the manuscript.

1. At the end of the Introduction section, it is better to indicate the organization of the study.

2. The contribution of this study should be highlighted at the end of the Introduction section.

3. In the Related work section, it is better to cite more references and discuss to motivate this study.

4. Regarding line 251-257, please re-organize them.

5. It is better to update "in this paper" as "in this study".

6. In line 303, "equation (15)" should be "equ.(15)".

7. In line 285, "The" should be "the".

8. It is better to ask a fluent English speaker to polish the language.

Reviewer 2 ·

Basic reporting

The authors proposed a Human-Computer interaction system based on background knowledge and emotion certainty. The proposed model simulates the natural human communication process to realize a more natural and intelligent human-computer interaction system. The experimental results show that, compared with the comparison model, the emotional friendliness and coherence of robots with background knowledge and emotional measurement can be effectively improved during human-computer interaction.

Experimental design

The experiments are designed from 4 aspects, i.e., Settings, Evaluation metrics, Experimental results and Experimental discussion, which seems reasonable.

Validity of the findings

This model achieves better results in both MAP and MRR objective ratings compared to the comparison models, as it takes into account both emotion and content friendliness, which can effectively improve the response accuracy.

Additional comments

The proposed model simulates the natural human communication process to realize a more natural and intelligent human-computer interaction system. The experimental results show that, compared with the comparison model, the emotional friendliness and coherence of robots with background knowledge and emotional measurement can be effectively improved during human-computer interaction. I have some major comments.
Abstract:
1. Please highlight the contributions and motivations. Currently, it is hard to catch them.
2. “The experimental results show that, compared with the comparison model, the emotional friendliness and coherence of robots with background knowledge and emotional measurement can be effectively improved during human-computer interaction.” It is best to give a quantitative description of the experimental results. For example, compared with the comparison model, how much the performance of the algorithm we proposed is improved specifically.
Introduction:
. Seem good. But it is best to give the specific contributions of this study and the differences from previous studies in the introduction section.
Related Work
4. “In [17], the authors proposed an emotion generation framework…”. I think this statement is not very good. If possible, please change it to “Rodríguez et al. proposed….”. Please check all sections.
Proposed method
5. In the section of Emotional friendliness, it is best to state the reason of “To ensure that the equation is meaningful, specifically h_l is not defined as 0. (Line 56-57)”
Experimental Studies
6. In the experimental setting part, the description of the comparison algorithm is best given in the form of a table. This makes the structure of the manuscript more compact. (Line 51-58)
Conclusions
7. “Human learning, living and working experiences are stored in the brain as memories in an associative manner, which can be regarded as personal background knowledge, and the process of human-human communication is the process of evoking locally associated background knowledge under the influence of emotional factors.” Please remove the background description in the conclusion section.
8. Please give the direction of subsequent research in the summary section.

Cite this review as

---

## Round 0.2 · Minor Revisions

One reviewer still has some minor comments regarding related works, conclusions, and the abstract. So please revise the manuscript carefully and resubmit it for further evaluation.

·

Basic reporting

no comment

Experimental design

no comment

Validity of the findings

no comment

Additional comments

no comment

Reviewer 2 ·

Basic reporting

The authors addressed my comment accordingly. However some minor comments should be considered

Experimental design

Please see details comments

Validity of the findings

Please see details comments

Additional comments

1. In lines 93-98, the author only discussed few related works. I highly suggest adding more.

2. The indication of future work is too simple. Please give more suggestions.

3. The abstract should be written in the past tense.

Cite this review as

---

## Round 0.3 · accepted · Accept

All reviewers suggest acceptance. Thanks for your contributions.

Reviewer 2 ·

Basic reporting

The authors have addressed all the comments

Experimental design

The authors have addressed all the comments

Validity of the findings

The authors have addressed all the comments

Cite this review as